# Head-Group Acylation of Chloroplast Membrane Lipids

**DOI:** 10.3390/molecules26051273

**Published:** 2021-02-26

**Authors:** Yu Song, Zolian S. Zoong Lwe, Pallikonda Arachchige Dona Bashanee Vinusha Wickramasinghe, Ruth Welti

**Affiliations:** 1Department of Biochemistry and Molecular Biophysics, Kansas State University, Manhattan, KS 66506, USA; songyu@ksu.edu (Y.S.); zolian@ksu.edu (Z.S.Z.L.); 2Kansas Lipidomics Research Center, Kansas State University, Manhattan, KS 66506, USA; vinuwickram@ksu.edu; 3Division of Biology, Kansas State University, Manhattan, KS 66506, USA

**Keywords:** *Arabidopsis thaliana*, plants, chloroplast lipids, plant stress, galactolipids, phosphatidylglycerol, head-group acylation, At2g42690, AGAP1

## Abstract

Head group-acylated chloroplast lipids were discovered in the 1960s, but interest was renewed about 15 years ago with the discovery of Arabidopsides E and G, acylated monogalactosyldiacylglycerols with oxidized fatty acyl chains originally identified in *Arabidopsis thaliana*. Since then, plant biologists have applied the power of mass spectrometry to identify additional oxidized and non-oxidized chloroplast lipids and quantify their levels in response to biotic and abiotic stresses. The enzyme responsible for the head-group acylation of chloroplast lipids was identified as a cytosolic protein closely associated with the chloroplast outer membrane and christened acylated galactolipid-associated phospholipase 1 (AGAP1). Despite many advances, critical questions remain about the biological functions of AGAP1 and its head group-acylated products.

## 1. Introduction

Chloroplasts are organelles of plant cells, responsible for photosynthesis, storage of photosynthetic products including starch, and synthesis of primary and secondary metabolites, including fatty acids. Chloroplasts are surrounded by inner and outer membranes (collectively called the envelope) and, inside, contain stacks of thylakoid membranes, where the light reactions of photosynthesis occur. Chloroplast membrane lipids include the non-phosphorous-containing glycolipids monogalactosyldiacylglycerol (MGDG), digalactosyldiacylglycerol (DGDG), and sulfoquinovosyldiacylglycerol (SQDG), the phospholipid, phosphatidylglycerol (PG), and, in the envelope, phosphatidylcholine (PC) [1,2]. In addition, several classes of chloroplast lipids are formed primarily when plants are under stress, including polygalactosylated diacylglycerols [3,4,5], glucuronosyldiacylglycerols [6], and head-group acylated MGDGs, DGDGs, and PGs [7,8,9,10]. Here, we discuss our current knowledge of the head group-acylated chloroplast lipids, acylated MGDG (acMGDG), acylated DGDG (acDGDG), and acylated PG (acPG).

## 2. Discovery of Head Group-Acylated Plastidic Lipids

In 1967, Heinz reported the discovery of a glycolipid more hydrophobic than MGDG, with a 1:1:3 molar ratio of glycerol, galactose, and fatty acids in spinach leaf homogenates [11]. Soon a similar lipid was discovered in wheat flour [12]. The lipid from both sources was shown to be MGDG with a third acyl group esterified to the carbon at the 6-position of the galactose [12,13] (Figure 1a).

Later work by Heinz and colleagues identified acDGDG, which was found to have a fatty acid esterified to the 6-carbon of the second galactose [14] (Figure 1b). AcPG was identified in oats and later also found in *Arabidopsis thaliana* leaves [9,15] (Figure 1c). While extra-plastidic head-group acylated polar lipids, such as *N*-acyl phosphatidylethanolamine, also have been identified in plants (as well as in animals), the focus of this review is the head-group acylated lipid classes of the plant chloroplast.

## 3. Acylated Monogalactosyldiacylglycerol (AcMGDG) Is Ubiquitous in Photosynthetic Tissue

AcMGDG species have been detected in green tissue of the major groups of land plants from non-vascular plants to angiosperms [16]. However, detected levels of acMGDG species after freeze-thawing vary considerably among plant species [16]. Outside the plant kingdom, only cyanobacteria and red algae have been reported to contain acMGDG species [17,18,19]. AcDGDG has been detected in spinach and *Arabidopsis* [14,20] and acPG has been found in oat seeds and *Arabidopsis* leaves [9,15,20], but the presence of acPG and acDGDG elsewhere in the plant kingdom has not been systematically investigated. Similarly, while acylated lipids or the ability to produce them has been detected in oat seeds [15] and daffodil flowers [21], the presence of acylated chloroplast lipids in non-photosynthetic tissues has not been systematically investigated.

## 4. Lipidomics Is Increasing Our Knowledge of Head Group-Acylated Lipid Molecular Species

The emergence of lipid analysis by electrospray ionization mass spectrometry, beginning around 2000, increased our ability to detect head group-acylated lipid molecular species present at low concentrations. It also provided the capability of analyzing larger numbers of biological samples. The structures of a limited number of head group-acylated lipid molecular species have been determined by nuclear magnetic resonance (NMR). As the number of detectable molecular species rapidly expanded with analysis by electrospray ionization mass spectrometry, accurate mass analysis of head group-acylated lipids and their fragments was used to identify the newly detected lipid molecular species at the level of chemical formula [10,20,22]. Ibrahim et al. used gas chromatography-mass spectrometry to carefully elucidate the structures of *Arabidopsis* fatty acids, then combined the fatty acid structural information with acylated lipid masses and fragmentation patterns detected by electrospray ionization mass spectrometry, to define acylated lipid structures [23]. In some cases, chromatography has been employed to provide separation before mass spectrometry [9,23,24], while in other cases, plant extracts have been infused directly to the mass spectrometer [10,20,22]. Lower-resolution mass spectrometers, such as triple quadrupole instruments, employing precursor scanning or multiple reaction monitoring, often have been used for quantification. Caution is necessary in identification of acMGDG species in direct-infusion triple quadrupole mass spectrometry methods, because there are acMGDG molecular species that are isobaric or near isobaric (have the same or nearly the same mass).

In 2006, Andersson et al. [7] purified and used mass spectrometry and NMR to identify an acMGDG containing the jasmonate precursors, oxophytodienoic acid (OPDA) and dinor-OPDA (dnOPDA) (Figure 2). This lipid was named Arabidopside E, joining a series of “Arabidopsides”, which are galactolipids that contain OPDA and dnOPDA [25,26,27]. Soon the same group discovered an all-OPDA version of acMGDG named Arabidopside G [8]. The observations of these *Arabidopsis* acMGDG molecular species containing multiple oxidized fatty acids fueled interest in a closer look at head group-acylated lipids in the chloroplast.

To date, more than 100 acylated chloroplast lipid molecular species have been observed by mass spectrometry, and, in many cases, the molecular species have been quantified in relation to internal standards [10,20,22,23,28]. While the internal standards used typically are not well-matched to head group-acylated lipids in terms of structure, quantitation in relation to internal standards allows comparison of the levels of acylated lipids among samples.

In general, acMGDG is much more common than acDGDG or acPG. The most common fatty acids found in the head group of acMGDG are 18:3, 16:0, and 16:3 in “16:3 plants” and 18:3 and 16:0 in “18:3 plants” [10,16,29]. Other detectable non-oxidized fatty acids are 18:2, 18:1, and 18:0. Besides OPDA and dnOPDA, oxidized acyl chains with anion formulas C_18_H_29_O_3_ (18:3;O), C_18_H_29_O_4_ (18:3;O2), and several others have been detected in *Arabidopsis* acMGDG [10,23]. However, the prevalence of oxidized fatty acids in chloroplast lipids varies widely among plant species, with OPDA-containing lipids having a somewhat limited phylogenetic distribution. OPDA-containing lipids are found in *Arabidopsis* and some other Brassicaceae, and have been identified in a few species outside this family, including *Melissa officinalis* (lemon balm), *Ipomoea tricolor* (Mexican morning glory), and *Cirsium arvense* (creeping thistle) [16,30,31,32]. In sum, oxidized lipids, particularly OPDA and dnOPDA, are common in *Arabidopsis*, and these species are present in head group-acylated lipid species. However, the presence of OPDA and/or dnOPDA in head group-acylated chloroplast lipids has not been investigated in other species that contain these fatty acids.

## 5. Varied Patterns of Acylated Chloroplast Lipids Are Induced by Environmental Stresses

Biotic and abiotic stresses result in dramatic fluctuations in chloroplast lipid levels, with head-group acylation being strongly enhanced by stress (Figure 3). Whereas wounding and freeze-thaw treatments have been shown to increase the level of acylated lipids in a number of plant species, most studies of stress effects on chloroplast lipid acylation have been done in *Arabidopsis thaliana* [7,8,10,16,20,22,23,28,33]. The pattern of acylated chloroplast lipid accumulation varies as a function of the particular stress applied to *Arabidopsis* plants. Mechanical wounding, pathogen infection, freezing, and heat stress all induce head-group acylation of chloroplast lipids. In the photosynthetic tissue of *Arabidopsis*, which produces more oxidized fatty acid during stress than many other plants, patterns of lipids produced during various stresses differ in the amount of lipids acylated and in the relative amount of oxidized vs. non-oxidized acylated lipids produced (Figure 3). Strikingly, in *Arabidopsis*, oxidized lipid species, which make up only a few percent of chloroplast lipids when intact plants are subjected to wounding or pathogen infection [10], are concentrated in head group-acylated molecular species (Figure 3).

When *Arabidopsis* (accession Columbia-0) is wounded, oxidation of membrane lipids is prominent, and levels of OPDA and dnOPDA increase in chloroplast lipids. Formation of acMGDGs, acDGDGs, and acPGs is induced by wounding [20]. Oxidized acMGDGs and non-oxidized acMGDGs are produced with different kinetics; oxidized acMGDGs are induced rapidly (levels are highest at 45 min and dropping 6 h after wounding), whereas non-oxidized acMGDGs are formed more slowly (levels are higher at 6 h than at 45 min) [10,20]. AcDGDG analogs of Arabidopsides E and G and an OPDA-containing acPG are also induced during wounding, following a pattern similar to that of the oxidized acMGDGs. The metabolic basis of the different kinetics of oxidized and non-oxidized lipid species in wounding stress is unknown.

Bacterial infection of plants can be either virulent or avirulent. The virulent pathogen *Pseudomonas syringae* pv *maculicola* (*Psm*) causes great damage to plants, but relatively slight changes to the lipidome [22]. Recognition of bacterial effector protein AvrRpm1 secreted by an avirulent strain of *P. syringae* leads to a hypersensitive response and dramatic lipid changes [7,8]. Transgenic AvrRpm1 production results in massive accumulation of Arabidopside E, Arabidopside G, and accumulation of two OPDA-containing acPGs [7,8,9]. Both oxidized acMGDGs and non-oxidized acMGDGs are greatly induced after recognition of AvrRpm1 [10]. In a virulent infection, the levels of Arabidopside E were only 8% as high as during the avirulent interaction [22]. Interestingly, analysis of a mutant in allene oxide synthase, an enzyme involved in OPDA synthesis, showed that in the absence of fatty acyl oxidation, the hypersensitive response occurring in avirulent infection still results in formation of acMGDGs, but mainly non-oxidized species [9].

Hot and cold temperatures also stress plants, resulting in the production of head-group acylated chloroplast lipids in *Arabidopsis* leaves. Sub-lethal freezing of *Arabidopsis* induces acMGDGs in a different accumulation pattern compared to wounding and pathogen infection (Figure 3). In freezing, levels of non-oxidized acMGDGs were significantly increased during the post-freezing period, while levels of oxidized acMGDGs were very low [10]. Strikingly, at 24 h after freezing, *Arabidopsis* plants that had been acclimated at 4 °C before freezing had 5-fold less acMGDGs than plants frozen without the 4 °C pre-treatment [10]. While, in general, increases of head group-acylated chloroplast lipids in heating stress are not as large as in avirulent bacterial interaction, wounding, or freezing stress, both non-oxidized and oxidized acMGDGs increase significantly when plants are subjected to heating stress [28,33].

## 6. AcMGDG Is Produced by Transacylation

Heinz’s early work demonstrated that the ability of spinach homogenates to produce acMGDG was destroyed by boiling, indicating that an enzyme was required for formation of the acylated lipid [11]. Incubating a protein extract of spinach leaves, at acidic pH, with purified MGDG resulted in synthesis of acMGDG, while MGDG levels dropped [34]. Data indicate that acMGDG is formed mainly by acyl transfer of a glycerol-linked fatty acid in DGDG to the galactose of MGDG, producing digalactosylmonoacylglycerol (DGMG) in addition to the acMGDG (Equation (1)). To a lesser extent, a glycerol-linked fatty acid can be transferred from one MGDG to the galactose of another MGDG, producing monogalactosylmonoacylglycerol (MGMG) in addition to the acMGDG, in a reaction that Heinz called dismutation (Equation (2)) [34]. There is only a slight positional specificity with regard to the fatty acyl chain that is transferred, with the fatty acyl chain in the sn-1 position being slightly favored [35]. Additionally, transfer of an acyl chain from a phospholipid donor has been observed, but phospholipids are less favored as donors compared to galactolipids [34].
DGDG + MGDG → acMGDG + DGMG(1)
2 MGDG → acMGDG + MGMG(2)

## 7. The Enzymatic Activity Catalyzing Head-Group Acylation of Chloroplast Lipids

Heinz, who initially identified acMGDG in spinach leaves, purified the MGDG acylating activity from leaves of broad beans (*Vicia faba*), determining that the enzymatic activity is highest at 40 °C and at pH 5.4 [36]. Nilsson and coworkers took up the quest to identify the enzyme responsible for formation of head group-acylated lipids in the chloroplast [16]. Based on their finding that acMGDG accumulated in large amounts in oat leaves after freeze-thaw treatment and considering the enzyme properties identified by Heinz, they purified the putative MGDG acyltransferase from oat leaves. A Basic Local Alignment Search Tool (BLAST) search of the amino acid sequence of the oat leaf protein against the *Arabidopsis* genome identified the gene *At2g42690*. Expression of At2g42690 in *E. coli* and in vitro analysis of the purified protein provided strong evidence for its function in catalyzing acMGDG formation. Given that At2g42690 belongs to a family of phospholipases and is tightly linked to the formation of head group-acylated galactolipids, the authors proposed the name acylated galactolipid-associated phospholipase 1 (AGAP1) for the enzyme responsible for acMGDG formation. Two *Arabidopsis* T-DNA insertion mutant lines, *agap1-1* and *agap1-2*, both failed to accumulate wild-type levels of acMGDG, and also acPG and acDGDG, after freeze-thaw treatment, indicating that AGAP1 is responsible for the acylation of all three lipids [16]. However, small amounts of acMGDG, acPG, and acDGDG are present in the *agap1* mutants and may be produced by related gene products.

## 8. Sub-Cellular Localization of Acylated Galactolipid-Associated Phospholipase 1 (AGAP1) and Head Group-Acylated Chloroplast Lipids

Heinz, Heemskerk, and colleagues found that the spinach MGDG-acylating activity was present in isolated outer chloroplast membranes from spinach leaves [37,38]. Based on the lack of a transit peptide and observed localization of a fluorescent tagged version of At2g42690 transiently expressed in tobacco, Nilsson et al. concluded that AGAP1 most likely resides in the plant cytosol and acts on galactolipids at the outer membrane of the chloroplast or on membrane fragments released by broken chloroplasts [16].

It is worth noting that in the Columbia-0 accession of *Arabidopsis* the most common molecular species, Arabidopsides E and G, which contain only OPDA or dnOPDA chains, account for 40–50% of the total head group-acylated chloroplast lipids produced in vivo after mechanical wounding or bacterial infection, although the percentage of OPDA in MGDG and DGDG is only approximately 1–2% (Figure 3) [10]. This may imply that the OPDA-containing galactolipids that likely serve as precursors to Arabidopsides E and G are sequestered in the location where head-group acylation occurs, away from the bulk non-oxidized MGDG and DGDG. Formation of OPDA from linolenic acid (18:3) in galactolipids occurs on the esterified chains (the intact galactolipid) rather than on free fatty acids [39]. A key enzyme leading to OPDA, allene oxide synthase, has been localized to the chloroplast envelope (inner and/or outer membranes) by several groups [40,41,42,43], although it has also been associated with the thylakoid membranes [44] and plastoglobules [45]. Based on the majority of these data, it could be speculated that the highly oxidized head-group acylated molecular species formed under stress are located in the envelope, and that the oxidation and acylation pathways are co-localized. Furthermore, it could be implied that the lipids undergoing oxidation and acylation reactions are not readily mixing with the bulk chloroplast lipids, either due to physical separation or substrate channeling among stress-responsive, lipid-metabolizing enzymes. Distinct localization might also affect catabolic enzyme access to head group-acylated lipids, leading to the observed shorter lives of oxidized vs. non-oxidized acylated molecular species [20], but all hypotheses related to the localization of acylation and acylated lipids remain to be tested.

## 9. Little Is Known about the Physical Properties of Head Group-Acylated Chloroplasts Lipids

Virtually nothing has been reported on the physical properties of chloroplast head group-acylated lipids. It is well-established that DGDG is a bilayer-forming lipid, while MGDG’s head group is small, and it tends to form inverted hexagonal phase, rather than a bilayer, unless its properties are modulated by the other bilayer components. These properties are similar to those of the phospholipids, PC, which is bilayer-forming, and phosphatidylethanolamine, which tends to form an inverted hexagonal phase. Although no studies have been undertaken on head group-acylated chloroplast lipids, there is analysis of the structure of *N*-acylated phosphatidylethanolamine [46]. This lipid’s third acyl chain is fully embedded in the membrane to a degree similar to the acyl chain in the *sn*2-position on the glycerol backbone. The *N*-acylation of phosphatidylethanolamine with a saturated or mono-unsaturated fatty acid results in only modest changes in the phase transition temperature of the membrane and interaction with other membrane components, compared to non-acylated phosphatidylethanolamines with the same acyl chains [47]. However, it seems possible that the situation could be different for acMGDG, given that the head-group acylation is likely to be either with a third polyunsaturated fatty acid, which has a high cross-sectional area, or even a third oxidized fatty acid, such as OPDA, which has a very high cross-sectional area, as well as a shorter length due to the presence of the cyclopentenone ring. Such head group acylation, along with the production of monoacyl lipid molecular species by AGAP1, could potentially affect bilayer stability. Galactolipids with multiple oxidized fatty acids also might have limited miscibility with other lipids in the membrane. Another possibility is that acylated molecular species such as Arabidopsides E and G could have significant water solubility that might allow them to move into the aqueous phase more readily than molecular species with non-oxidized fatty acids. At this juncture, this is speculation.

## 10. The Biological Functions of AGAP1 and Chloroplast Lipid Head-Group Acylation Remain Elusive

Despite ample evidence that acylated galactolipids (oxidized and non-oxidized) accumulate in response to biotic and abiotic stresses, the biological functions of AGAP1 and head group-acylated chloroplast lipids remain elusive [7,8,10,16]. Both *agap1* mutants appear to be unaffected by their defect in acylated lipid production under standard growth conditions. Here we consider three hypotheses about the function of AGAP1 and the lipids it acylates.

**Hypothesis** **1.**
*High levels of head group-acylated lipids produced in pathogen response or wounding help plants survive the stresses. Arabidopside E has been demonstrated to have slight bactericidal activity against Pseudomonas syringae [7], and both Arabidopsides E and G somewhat reduce growth of the necrotrophic fungus Botrytis cinerea, while free OPDA has no effect [8]. Despite the anti-pathogen effects of the tested acMGDGs, Nilsson et al. [16] found that the agap1-1 mutant did not differ much from wild type in its ability to produce a hypersensitive response to P. syringae expressing an avirulent effector or in the ability of the bacteria to grow in their leaves. The authors concluded that AGAP1 and head group-acylated lipids formed during bacterial infection likely do not have a major role in the outcome of plants responding to pathogens via the hypersensitive response. The agap1-1 mutant and wild-type plants were also similar in their ability to support the feeding and growth of cotton leafworms (Spodoptera littoralis).*


**Hypothesis** **2.**
*The function of head group-acylated galactolipids containing OPDA and dnOPDA is to serve as phytohormone reservoirs that could result in faster or larger responses in jasmonates if the plants encounter a second stimulus. Vu and coworkers tested this hypothesis by rewounding Arabidopsis leaves 24 or 48 h after a first wounding [10]. Although the levels of acMGDGs were higher after the second wounding event, the levels of free OPDA and jasmonic acid were not significantly higher nor did they appear more quickly after rewounding.*


**Hypothesis** **3.**
*AGAP1 and head group-acylated chloroplast lipids play a role in sequestration of acyl chains during an enhanced period of acyl-chain turnover caused by biotic or abiotic stress. This hypothesis suggests that AGAP1 might play a role analogous to that proposed for another transacylase, phospholipid:diacylglycerol acyltransferase 1 (PDAT1), which transfers an acyl chain from PC to diacylglycerol to produce triacylglycerol and lysoPC. Mueller et al. [33] demonstrated that Arabidopsis PDAT1 contributes to basal thermotolerance by sequestering displaced fatty acids in triacylglycerol during heat stress. Thus, one possibility is that the ability of AGAP1′s transacylation activity to remove acyl chains from the membrane under stress conditions helps plants during or after stress. This remains to be tested.*


Nilsson et al. [16] point out that the presence of head group-acylated chloroplast lipids across the plant kingdom and their maintenance through considerable evolutionary time suggests that they have a biological role. They also suggest that the role of AGAP1 and head group-acylated chloroplast lipids likely lies in realm of response to biotic and abiotic stresses that damage the chloroplast. We concur and suggest that characterization of the physical properties of the head group-acylated lipids, further work on their subcellular location, and continued analysis of *AGAP1* mutants may shed light on the role of AGAP1 and the ubiquitous modification of chloroplast lipids by head-group acylation.

## Figures and Tables

**Figure 1 molecules-26-01273-f001:**
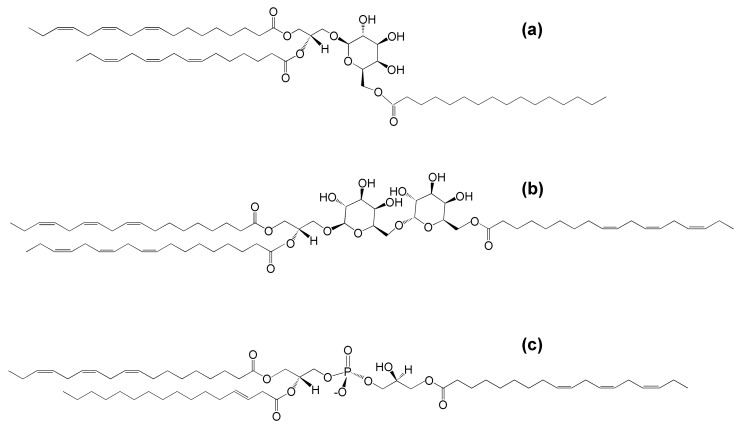
The molecular structures of (**a**) acylated monogalactosyldiacylglycerol (acMGDG), i.e., 1,2-di-*O*-acyl-3-*O*-(6′-*O*-acyl-β-d-galactopyranosyl)-*sn*-glycerol (*sn*1 = 18:3^∆9,12,15^, *sn*2 = 16:3^∆7,10,13^, 6′ = 16:0), (**b**) acylated digalactosyldiacylglycerol (acDGDG), i.e., 1,2-di-*O*-acyl-3-*O*-[6′-*O*-(6″-*O*-acyl-α-d-galactopyranosyl)-β-d-galactopyranosyl)]-*sn*-glycerol (*sn*1 = *sn*2 = 6″ = 18:3^∆9,12,15^) and (**c**) acylated phosphatidylglycerol (acPG), i.e., 1-,2,di-*O*-acyl-*sn*-glycero-3-phospho-(3-acyl)-1-glycerol (*sn*1 = 18:3^∆9,12,15^, *sn*2 = 16:1^∆3trans^, 3′ = 18:3^∆9,12,15^).

**Figure 2 molecules-26-01273-f002:**
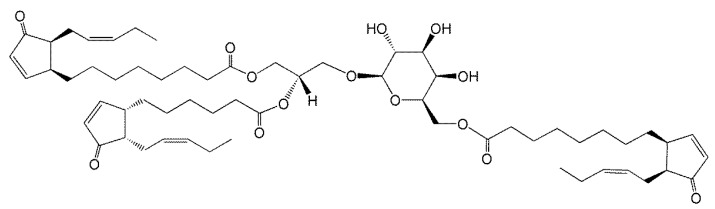
Arabidopside E, 1-*O*-OPDA,2-*O*-dnOPDA-3-*O*-(6′-*O*-OPDA-β-d-galactopyranosyl)-*sn*-glycerol, where OPDA and dnOPDA are abbreviations for oxophytodienoic acid and dinor-oxophytodienoic acid, respectively.

**Figure 3 molecules-26-01273-f003:**
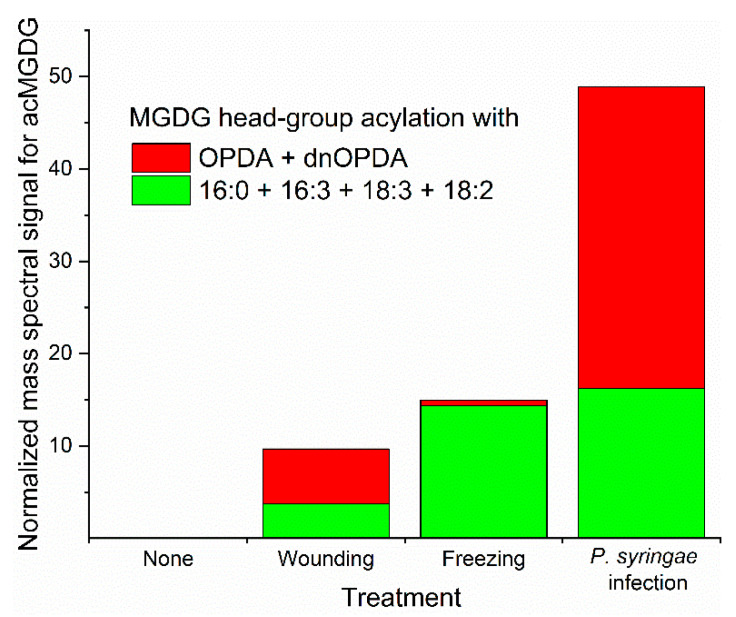
*Arabidopsis thaliana* accession Columbia-0 produces different acMGDG amounts and different molecular species when subjected to different stresses. Relative amounts of acMGDG head-group acylation by non-oxidized (16:0 + 16:3 + 18:3 + 18:2) (green) and oxidized (OPDA + dnOPDA) (red) on monogalactosyldiacylglycerol (MGDG) as a function of stress treatment. “*Pseudomonas syringae* infected” represents an avirulent infection at 24 h. Freezing was 2 h at −8 °C without acclimation and sampling after a 24 h recovery at 21 °C. Wounding was with a hemostat across the midvein and sampling 6 h later. Data are from reference 10.

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
