# Peer review of "Head-Group Acylation of Chloroplast Membrane Lipids"

_molecules, 2021, doi:10.3390/molecules26051273_

Round 1
Reviewer 1 Report
The manuscript by Song and colleagues is a comprehensive review of the field of head group acylated chloroplast lipids. Overall, I find it well written, interesting and very comprehensive. I have only a few suggestions for improvements that the authors may want to consider:
Abstract, line 12: I suggest rephrasing from “Arabidopsis acyalted… “ to “acylated monogalactosyldiacylglycerols with oxidized fatty acyl chains in Arabidopsis.”
Section 2, line 40: “lipids” change to “lipid”
Title section 3: I suggest delete “found” in title
Section 4, line 102: This must be the head group acylated fatty acid that is referred to, not the diacylglycerol?
Section 5, line 130: Perhaps clarify that “a few percent” refers to unstressed material, following freeze thaw or something similarly brutal, Arabidopsis converts a large proportion of it’s MGDG into (dn)OPDA containing species.
Section 5, line 148: Change “under wounding” to “during wounding”.
Section 5, line 159: It’s debatable whether “infection” is a proper term for the avirulent interaction since avirulence implies lack of established infection. Perhaps just change “infection” to “interaction”.
Section 5, line 171: “infection” remove or change to “interaction”.
Section 7, line 205: It should be noted that the agap-mutants still produce detectable amounts of head group acylated lipids, there might thus be some redundancy and there are several similar genes in the Arabidopsis genome.
Section 8, line 219: Again it should be noted that the few percents apply to an unstressed condition. More generally, it has been shown the when Arabidopsis leaf tissue is rapidly disrupted, a large proportion of the MGDG is rapidly converted to (dn)OPDA containing MGDG and thereafter there is a slower reshuffling of the (dn)OPDA into head group acylated species. The initial step is also shown to be dependent on LOX2 (See Nilsson 2012 (10.1016/j.febslet.2012.06.010) and Nilsson 2016 (10.1093/jxb/erw278)).
Section 9, line 243: There are studies underway from the group of Marie Laure Fauconnier regarding the physical protperties of Arabidopsides. No “real” publications yet, but there may be interesting things coming out soon. Try for instance this link: https://scholar.google.com/scholar?oi=bibs&cluster=16755014822048412531&btnI=1&hl=fr
Section 9, lines 258-261: That the head group acylated arabidopsides would have a high water solubility sounds to me like a quite unlikely suggestion. I have no particular data to back this up, but given that it behaves as less polar than most other membrane lipids in both straight and reverse phase chromatography might be an indication.
Section 10, line 277: Pseudomonas mainly grows “in” the leaves rather than “on”.
Author Response
Reviewer 1:
Thank you for your comments. We appreciate them and provide a point-by-point response below.
The manuscript by Song and colleagues is a comprehensive review of the field of head group acylated chloroplast lipids. Overall, I find it well written, interesting and very comprehensive. I have only a few suggestions for improvements that the authors may want to consider:
Abstract, line 12: I suggest rephrasing from “Arabidopsis acyalted… “ to “acylated monogalactosyldiacylglycerols with oxidized fatty acyl chains in Arabidopsis.”
Response: We have moved the plant identifier, adding “originally identified in Arabidopsis thaliana” at the end of the sentence.
Section 2, line 40: “lipids” change to “lipid”
Response: This has been changed and the verb changed to agree with the subject.
Title section 3: I suggest delete “found” in title
Response: This was deleted and “ubiquitous” changed from adverb to adjective form.
Section 4, line 102: This must be the head group acylated fatty acid that is referred to, not the diacylglycerol?
Response: We have changed it accordingly.
Section 5, line 130: Perhaps clarify that “a few percent” refers to unstressed material, following freeze thaw or something similarly brutal, Arabidopsis converts a large proportion of it’s MGDG into (dn)OPDA containing species.
Response: This statement refers to data from reference 10. We clarified to indicate that the conditions are “sub-lethal freezing (2 h at -8° C)”.
Section 5, line 148: Change “under wounding” to “during wounding”.
Response: Thanks!
Section 5, line 159: It’s debatable whether “infection” is a proper term for the avirulent interaction since avirulence implies lack of established infection. Perhaps just change “infection” to “interaction”.
Response: This was changed as suggested.
Section 5, line 171: “infection” remove or change to “interaction”.
Response: This was changed as suggested.
Section 7, line 205: It should be noted that the agap-mutants still produce detectable amounts of head group acylated lipids, there might thus be some redundancy and there are several similar genes in the Arabidopsis genome.
Response: We noted that small amounts of acylated chloroplast lipids remain in the agap1 mutants and suggested that there could be other gene products producing the acylated lipids. Thanks for this suggestion.
Section 8, line 219: Again it should be noted that the few percents apply to an unstressed condition. More generally, it has been shown the when Arabidopsis leaf tissue is rapidly disrupted, a large proportion of the MGDG is rapidly converted to (dn)OPDA containing MGDG and thereafter there is a slower reshuffling of the (dn)OPDA into head group acylated species. The initial step is also shown to be dependent on LOX2 (See Nilsson 2012 (10.1016/j.febslet.2012.06.010) and Nilsson 2016 (10.1093/jxb/erw278)).
Response: Again, this refers to data from reference 10. The mechanical wounding was at a level at which the plants can recover completely in 24 h with no mark left on the leaf. This is a different situation than rapid disruption of leaf tissue. However, it’s clear that in this type of wounding (dn)OPDA is concentrated in acMGDG.
Section 9, line 243: There are studies underway from the group of Marie Laure Fauconnier regarding the physical protperties of Arabidopsides. No “real” publications yet, but there may be interesting things coming out soon. Try for instance this link: https://scholar.google.com/scholar?oi=bibs&cluster=16755014822048412531&btnI=1&hl=fr
Response: That looks interesting. I’m happy to know about it, but we won’t mention it here, because it looks like characterization of Arabidopside E and G is not described yet.
Section 9, lines 258-261: That the head group acylated arabidopsides would have a high water solubility sounds to me like a quite unlikely suggestion. I have no particular data to back this up, but given that it behaves as less polar than most other membrane lipids in both straight and reverse phase chromatography might be an indication.
Response: You are probably right, but we plan to leave this suggestion, which was inspired by the “lipid whisker” model in animal cells. And as we already noted, it’s speculation.
Section 10, line 277: Pseudomonas mainly grows “in” the leaves rather than “on”.
Response: Thanks. We changed that.
Your comments are appreciated.
Reviewer 2 Report
This review from the Welti lab group was quite well written, and made the somewhat difficult subject material quite accessible. The subject of head group acylation in the chloroplast is an interesting puzzle, and the authors do an excellent job of providing a comprehensive, balanced view of the topic-- from the historical discovery the more recent identification of the gene responsible. Several plausible hypotheses are put forward for the function of this headgroup acylation, and the scavenging of acyl groups for membrane remodeling during stress seems quite reasonable. The authors identify key questions for further work, which is always much more satisfying than a simple recapitulation of past research results.
I recognize that the focus here is on acylation of head groups on chloroplast lipids, but a mention of the acylation of PE that occurs outside of chloroplasts could also support a general cellular mechanism for the scavenging of fatty acids (see for example Rawlyer and Braendle, 2001, Plant Physiol) -- this process happens in plant and animal cells and results in the formation of N-acylPE. Although N-acylPE has become of most interest recently as the precursor for acylethanolamide signaling lipids like anandamide, there is literature suggesting these lipids accumulate in response to cellular injury as well. The authors may or may not wish to add a sentence or two in this regard.
In all, this excellent article is informative and timely with the necessary figures to illustrate the key types of lipid structures. The review is of appropriate scope and depth and should serve as a resource for future lipid scientists intrigued by these unusual membrane lipids. I have no suggestions for improvement.
Author Response
Reviewer 2:
This review from the Welti lab group was quite well written, and made the somewhat difficult subject material quite accessible. The subject of head group acylation in the chloroplast is an interesting puzzle, and the authors do an excellent job of providing a comprehensive, balanced view of the topic-- from the historical discovery the more recent identification of the gene responsible. Several plausible hypotheses are put forward for the function of this headgroup acylation, and the scavenging of acyl groups for membrane remodeling during stress seems quite reasonable. The authors identify key questions for further work, which is always much more satisfying than a simple recapitulation of past research results.
I recognize that the focus here is on acylation of head groups on chloroplast lipids, but a mention of the acylation of PE that occurs outside of chloroplasts could also support a general cellular mechanism for the scavenging of fatty acids (see for example Rawlyer and Braendle, 2001, Plant Physiol) -- this process happens in plant and animal cells and results in the formation of N-acylPE. Although N-acylPE has become of most interest recently as the precursor for acylethanolamide signaling lipids like anandamide, there is literature suggesting these lipids accumulate in response to cellular injury as well. The authors may or may not wish to add a sentence or two in this regard.
Response: This is an interesting thought, but we think that it would be outside of the scope of the review to add in significant discussion of the function of NAPE. We did add the following sentence on lines 46-48: “While extra-plastidic head-group acylated polar lipids, such as N-acyl phosphatidylethanolamine, also have been identified in plants (as well as in animals), the focus of this review is the head-group acylated lipid classes of the plant chloroplast.”
In all, this excellent article is informative and timely with the necessary figures to illustrate the key types of lipid structures. The review is of appropriate scope and depth and should serve as a resource for future lipid scientists intrigued by these unusual membrane lipids. I have no suggestions for improvement.
Response: We appreciate your generous comments.